# Hepatitis B Virus Research in South Africa

**DOI:** 10.3390/v14091939

**Published:** 2022-08-31

**Authors:** Mohube B. Maepa, Abdullah Ely, Anna Kramvis, Kristie Bloom, Kubendran Naidoo, Omphile E. Simani, Tongai G. Maponga, Patrick Arbuthnot

**Affiliations:** 1Wits/SAMRC Antiviral Gene Therapy Research Unit, Faculty of Health Sciences, Infectious Diseases and Oncology Research Institute (IDORI), University of the Witwatersrand, Johannesburg 2000, South Africa; 2Hepatitis Diversity Research Unit, Department of Internal Medicine, Faculty of Health Sciences, School of Clinical Medicine, University of the Witwatersrand, Johannesburg 2000, South Africa; 3National Health Laboratory Service, Johannesburg 2000, South Africa; 4HIV and Hepatitis Research Unit, Department of Virology, Sefako Makgatho Health Sciences University, Pretoria 0204, South Africa; 5Division of Medical Virology, Faculty of Medicine and Health Sciences, Stellenbosch University, Cape Town 7602, South Africa

**Keywords:** HBV, HBV/HIV co-infection, occult HBV infection, vaccination, gene therapy

## Abstract

Despite being vaccine-preventable, hepatitis B virus (HBV) infection remains the seventh leading cause of mortality in the world. In South Africa (SA), over 1.9 million people are chronically infected with HBV, and 70% of all Black chronic carriers are infected with HBV subgenotype A1. The virus remains a significant burden on public health in SA despite the introduction of an infant immunization program implemented in 1995 and the availability of effective treatment for chronic HBV infection. In addition, the high prevalence of HIV infection amplifies HBV replication, predisposes patients to chronicity, and complicates management of the infection. HBV research has made significant progress leading to better understanding of HBV epidemiology and management challenges in the SA context. This has led to recent revision of the national HBV infection management guidelines. Research on developing new vaccines and therapies is underway and progress has been made with designing potentially curative gene therapies against HBV. This review summarizes research carried out in SA on HBV molecular biology, epidemiology, treatment, and vaccination strategies.

## 1. Introduction

In 2019, an estimated 296 million people were living with chronic hepatitis B virus (HBV) infection (CHB), and the virus was responsible for approximately 1.5 million new infections and 887,000 deaths world-wide [1,2]. The prevalence of HBV infection is particularly high in sub-Saharan Africa, with about 60 million people chronically infected. Owing to inadequate screening and disease surveillance, HBV infection incidences in SA are underestimated and CHB remains neglected. HBV is endemic in SA with the highest rates in adults, especially those who are co-infected with HIV [3]. It is estimated that about 7.4% of the world’s population infected with HIV is also chronically infected with HBV. More than 70% of these HBV/HIV co-infected individuals reside in sub-Saharan Africa. SA has the largest number of HIV infections in the world with over 7.5 million people living with the virus [4].

The HBV genome comprise the *polymerase*, *surface*, *core*, and *X* overlapping open reading frames (ORFs) encoding structural, enzymatic, and regulatory proteins. The partly double stranded relaxed circular DNA (rcDNA) minus strand comprises 3200 nucleotides, and the shorter plus strand is of variable length. Regulatory elements include four promoter sequences (pS1, pS2, pC, pX) and two enhancer elements (Enh1 and Enh2) [5]. At the start of infection, glycosaminoglycans mediate clustering of viral particles on the surface of hepatocytes. The sodium taurocholate co-transporting polypeptide (NTCP) receptor on the surface of hepatocytes initiates entry of the virus into host cells [6]. With the help of the epidermal growth factor receptor, the virion enters the cell using a poorly understood mechanism. After entering hepatocytes, the nucleocapsid is directed to the nucleus where the rcDNA is released and undergoes ‘DNA repair’ to form covalently closed circular DNA (cccDNA). This stable construct, also referred to as the HBV minichromosome, allows the virus to persist in infected cells. cccDNA serves as the template for transcription of the pre-genomic RNA (pgRNA) as well as other major viral mRNAs. Transcripts are exported to the cytoplasm where they are translated into viral proteins, including hepatitis B surface antigen (HBsAg), hepatitis B core antigen (HBcAg), hepatitis B e antigen (HBeAg), DNA polymerase (pol), and the hepatitis B X protein (HBx). HBsAg can also be produced following transcription of viral DNA that has been integrated into the host genome. Viral capsids are made up of HBcAg proteins that are assembled into icosahedrons containing pgRNA and viral polymerase. Within the nucleocapsid, pol mediates rcDNA synthesis by reverse transcribing pgRNA to form the minus DNA strand, which serves as a template for plus strand synthesis. HBV capsids undergo maturation and are coated with HBsAg embedded envelope before they are released from hepatocytes. rcDNA-containing capsids may also be recycled to the nucleus to maintain cccDNA pools [5].

HBV is classified into nine genotypes, A–I [7,8,9,10], based on an intergroup divergence of >7.5% across the complete genomes [8,10]. A putative 10th genotype, “J”, has been isolated from a single case in Japan [11]. Genotypes A–D, F, H, and I are classified further into at least 35 subgenotypes. This is based on intergroup nucleotide divergence of between 4% and 8% across complete genomes [7,8,10]. The genotypes, and in some cases, the subgenotypes, have distinct global and local geographical distributions [8,10]. It was first recognized that HBV genotypes can be divided into subgenotypes, when the pre-S2/S region of South African isolates was sequenced [12]. Existence of subgenotypes was subsequently confirmed by further sequencing of pre-S1/pre-S2/S and complete genomes, leading to identification of subgenotype A1 [13,14]. In addition to distinctive sequence characteristics, subgenotype A1 differs from other (sub)genotypes in epidemiological and clinical features [15]. Subgenotype A1 is found in 70% of Black South African carriers of HBV and is also found in southeastern Africa [16,17] whereas subgenotype A2 circulates mainly outside Africa [13].

In SA, the hepatitis B vaccine was introduced into the Expanded Programme on Immunisation (EPI) in 1995. The vaccination regimen involves administration of the HBsAg-based vaccine at 6, 10, and 14 weeks post-natally, then a booster at 18 months of age, all as part of a multivalent vaccine dose [18]. Limitations of the program are that it excludes those born before 1995 and there is no birth dose. The latter allows significant numbers of HBV infections in SA to occur perinatally following transmission to infants from viremic HBV-infected mothers. Nevertheless, introduction of the HBV vaccine as part of childhood immunization has significantly decreased infection rates in some parts of SA [19,20]. Recent recommendations of the South African Viral Hepatitis Guidelines are for the introduction of a catch-up vaccination program, maternal HBV screening, and HBV mother-to-child prevention with vaccination at birth [18]. However, these recommendations are yet to be implemented. Novel HBV prevention strategies that are being explored include mRNA and viral vector-based vaccines.

The gold standard for HBV infection diagnosis is the detection of HBsAg in serum or plasma using enzyme-linked immunosorbent assays (ELISAs) or chemiluminescence immunoassays [21]. Acute HBV infection is characterized by presence of HBsAg for less than six months, and CHB is defined as the persistence of HBsAg for at least six months. The earliest serological markers to appear following HBV exposure are the HBsAg and antibodies to HBcAg (anti-HBc) [22]. Although poorly understood and commonly defined as another phase of CHB, occult HBV infection (OBI) is characterized by a lack of HBsAg positivity but presence of HBV DNA in the liver, with or without HBV DNA in the serum [23,24]. In SA, OBI prevalence ranges between 5.4% and 23%, depending on the study population [25,26,27].

Management of acute HBV infection is mainly supportive because more than 95% of immunocompetent individuals recover from the infection. First line treatment for CHB includes PEGylated interferon and nucleoside/nucleotide analogs such as tenofovir, entecavir, and lamivudine. The end goals of treating CHB are to prevent progression to cirrhosis; hepatic decompensation and hepatocellular carcinoma (HCC) in cirrhotic patients; and improvement of synthetic function in those with decompensated liver disease secondary to CHB. The loss of HBsAg coupled with seroconversion to anti-HBs is also a desired goal of treating CHB. Tenofovir in the form of tenofovir disoproxil fumarate (TDF) is most used as first line for treatment of CHB in South Africa. Treatment using TDF is long term and is frequently given indefinitely because of the risk of reactivation infection when therapy is withdrawn or terminated [18]. Although treatment regimens including these drugs may achieve a functional cure, they rarely eliminate the cccDNA from the infected hepatocytes. Functional cure from CHB is characterized by loss of HBsAg with reduced risk of hepatocellular carcinoma (HCC), cirrhosis, and liver failure. However, failure to eliminate cccDNA typically results in relapse following discontinuation of therapy. Treatment with lamivudine may result in emergence of viral escape mutants. Studies using samples from lamivudine-treated or -naïve SA patients with CHB reported reverse transcriptase gene mutations in 75.6% of isolates from patients receiving lamivudine, and 38% of cases were associated with drug resistance and treatment failure [28]. Interestingly, lamivudine resistance mutations were also detected in 37% of lamivudine-naïve patients [28,29] and in HIV/HBV co-infected individuals [30,31]. The resistance-associated mutations detected from lamivudine-naïve patients were likely to be polymorphisms of the wild-type viral sequence [32]. Hence, development of resistance to nucleoside/nucleotide analogs with a low genetic barrier to resistance such as lamivudine poses a challenge to anti-HBV therapy. New strategies to treat HBV infection include gene therapy approaches [33,34,35,36,37]. This review focuses on South African research that is aimed at understanding the biology and epidemiology of HBV, HBV/HIV coinfection, OBI, and new strategies to improve the prevention and treatment of HBV infection.

## 2. Molecular and Functional Characterization of HBV Subgenotype A1: The Viral Strain Prevailing in South Africa

Sequencing of many A1 isolates led to the conclusion that this subgenotype is endemic to Africa and has a long evolutionary history on the continent [13,38]. Outside of Africa, A1 is confined to areas where there has been a history of recent migration from Africa [39,40,41]. A case-control study showed that Africans infected with subgenotype A1 have a 4.5 times higher risk of developing liver cancer than those infected with other (sub)genotypes and these patients develop cancer at an earlier age [42]. The high hepatocarcinogenic potential may be a result of the subgenotype’s distinctive sequence characteristics, which are responsible for high HBeAg-negativity [15,43,44] and low HBV DNA levels in carriers [15]. Research within the Hepatitis Diversity Research Unit, Johannesburg, has extensively characterized A1 variants at the molecular and functional level.

Using extensive bioinformatic analyses, variations in the basic core promoter (BCP) and precore (pre-C) region positively associated with subgenotype A1 were identified (Table 1) [45]. Nucleotide 1888A can interfere with initiation at the downstream 1901 core AUG, thereby decreasing core protein translation [46]. On the other hand, mutations in the BCP/Pre-C can affect HBeAg expression at the transcriptional (A1762T/G1764A), translational (GCAC to TCAT at 1809–1812 from the *Eco*RI site) and post-translational levels (G1862T) [47,48]. Although not unique to subgenotype A1, A1762T/G1764A is frequently found in subgenotype A1 HBV isolated from patients with HCC [49,50,51]. This mutant results in decreased HBeAg expression [49] and has been shown to be a risk factor for development of HCC [52]. TCAT at 1809–1812, found in the Kozak sequence, is characteristic of subgenotype A1 and causes leaky ribosomal scanning to affect translation of HBeAg [53].

Almost exclusive to subgenotype A1 [45,47], G1862T is more frequent in HBV from HBeAg-negative than in HBeAg-positive South African carriers [56,57]. The sequence is also often found in HBV from HCC tumors, but not in adjacent non-tumorous liver tissue [57]. The 1862 G to T transversion causes a valine to phenylalanine substitution at the -3 position relative to the signal peptide cleavage site of the precursor protein. The aromatic ring of the phenylalanine interferes with the signal peptide function and processes necessary for the maturation of the precursor to HBeAg [58]. After introducing the G1862T mutation into wild-type genotype D sequences, a 54% reduction in secretion of HBeAg was observed [59]. Following transfection with a replication-competent subgenotype A1 clone, G1862T also diminished HBeAg expression, albeit to a lower degree (22%) [60]. HBeAg precursor protein accumulated in the endoplasmic reticulum (ER) and endoplasmic reticulum Golgi intermediate compartment (ERGIC) [60]. This accumulation triggered an earlier activation of the three unfolded protein response (UPR) pathways, leading to increased ER stress without increasing apoptosis [60]. HBeAg is immunomodulatory and reduced concentrations redirect the immune response to HBV-infected hepatocytes. Together with increased ER stress, this can cause liver damage and contribute to the higher hepatocarcinogenic potential of subgenotype A1 [47]. These BCP/Pre-C mutations either individually or in combination result in diminished circulating HBeAg.

When HBV sequences isolated from SA with and without HCC were compared, the previously reported 1762T/1762A substitution was observed together with cancer-associated mutations in the viral pre-S region. This includes pre-S2 deletion, mutations in the pre-S2 initiation codon, and pre-S2F22L, which may be associated with viral immune escape (Table 1) [50]. Similar observations were made in HBV strains isolated from Indian [40] and Kenyan [17] HCC patients infected with subgenotype A1. A link between these mutants and the pathogenesis of HCC has been shown in experimental and clinical studies [61,62]. Pre-S deletion mutants, identical to those isolated from HCC patients, also occur in subgenotype A1 isolates from HIV-infected individuals [30]. The A1762T/G1764A and pre-S deletions occurred more frequently in HBV/HIV co-infected subjects than in HBV mono-infected individuals [54].

To characterize subgenotype A1 further, replication-competent plasmids of HBV subgenotypes A1, A2, and D3, with authentic endogenous promoters were analyzed [63]. Replication of the three subgenotypes was compared after transfection. Subgenotype A1 expressed the lowest levels of precore/core precursor [44]. HBeAg, core expression, and replicative activity were lowest both in vitro and in vivo [48,64]. There was increased retention of viral proteins in the secretory pathway, increased ER stress, and earlier and prolonged activation of UPR in cells transfected with subgenotype A1 [44]. When liver cells were transfected with replication competent plasmids of subgenotype A1, precore protein expression was affected by absence of core, which also influenced HBsAg expression, suggesting an interrelationship between precore proteins, HBcAg, and HBsAg [65]. Incorporating greater-than-genome-length subgenotype A1 sequences into recombinant adeno-associated viruses (AAVs), HBV replication was demonstrated in vitro in hepatoma cells and in mice [66]. These infection experiments resulted in lower HBV gene expression of subgenotype A1 compared to D3, mirroring observations in patients and results from transfection of hepatoma cells with replication-competent plasmid clones [15,43,44]. The modulatory role of HBeAg and its precursors, as well as the lower replicative activity of subgenotype A1, may be important for immune escape and viral persistence and ultimately contribute to development of HCC.

## 3. HBV/HIV Co-Infection and Treatment

The high numbers of HIV and HBV infections in SA have led to the country being described as having a syndemic of HIV and HBV [67]. The prevalence of HBV infection is generally similar in HIV-infected and HIV-uninfected adults because most infections of hepatitis B are acquired in early childhood, before the acquisition of HIV in adulthood [68]. HBV/HIV co-infection is known to increase HBV replication rates, delay HBeAg antigen seroconversion and increase likelihood of developing CHB [69].

With implementation of antiretroviral therapy (ART) for HIV, the incidence of HBV-related liver disease and mortality has also increased [70]. This is because HBV and HIV-coinfected persons are now living longer and dying from HBV-related cirrhosis and hepatocellular carcinoma [70,71]. Furthermore, data suggest that HIV hastens the progression of CHB to HCC, with Africa-based studies reporting that HCC develops a decade earlier in HIV/HBV co-infected patients than in people carrying only HBV [72,73]. In addition, survival following HCC diagnosis in HIV/HBV co-infected patients appears worse compared to individuals carrying HBV alone [72]. Median survival for co-infected individuals with HCC was 81 days compared to 181 days for patients infected with only HBV. Unfortunately, these studies are limited by low sample numbers, and the impact of HIV ART on progression to HCC is not yet conclusive. Since 2017, SA has implemented a test-and-treat policy for HIV infection. This ensures that HIV-infected patients are promptly treated without a need for measurement of CD4 cell counts [74]. The benefits of ART in those who are HIV-infected are clear. There are reductions in mortality and morbidity from HIV-related conditions and improvement in life expectancy [75]. In contrast, treatment of HBV is not a straightforward process because of the complex clinical staging and treatment eligibility guidelines [76]. Treatment in those infected with HBV alone is only indicated for patients with any of the following: advanced fibrosis or cirrhosis, acute liver failure to prevent further hepatocyte death, or those receiving chemotherapy, rituximab, or immunosuppressive agents [18]. The benefits of HIV treatment accrue to those patients with HIV/HBV co-infection because the current first-line ART in SA includes the drugs TDF and lamivudine or emtricitabine, which are also active against HBV [77]. For HIV/HBV co-infected patients, these drugs are provided as a fixed drug combination for adults, adolescents, and children more than 3 years of age [78]. The use of TDF is particularly desirable because of the high barrier against drug-resistant HBV when compared to the previous regimens that contained lamivudine as the only HBV-active drug [77,79]. Studies on SA patients have shown a clinical advantage in co-infected patients on ART compared to HBV mono-infected patients who do not always get antiviral therapy even when it may be clinically indicated. These treated HIV/HBV co-infected patients have improved liver fibrosis scores and lower hepatitis B viral loads, although they still have higher serum prevalence of HBeAg compared to individuals infected with HBV alone [80,81]. Furthermore, in European cohorts, treatment of non-cirrhotic HIV/HBV co-infected patients led to decreased incidence of HCC [82]. Such studies need to be undertaken in sub-Saharan Africa.

The treatment of HBV in HBV/HIV co-infected patients provides a unique opportunity to study the emergence of variants that are resistant to currently available antivirals, particularly to TDF. There are emerging data from SA showing that not all the patients that are on ART achieve HBV suppression despite adherence to regimens that contain TDF [83,84]. Mutations that can explain this non-suppression of HBV, which frequently occurs in a background of suppressed HIV replication, have not yet been defined, but non-adherence is excluded as the cause of persistent circulating HBV DNA.

## 4. Occult HBV Infection

Patients with OBI usually have low levels of circulating HBV DNA with antibodies to HBcAg as the only serological marker of infection [23,85]. In some cases, anti-HBsAg may be present as has been reported in Mozambican patients [86]. Although highly viremic (>350 copies/mL) patients have been identified where anti-HBc was the only marker of HBV infection, OBI is generally characterized by HBV DNA concentrations of less than 200 copies/mL [23,87]. Treatment is not recommended for patients with OBI in SA, however, patients should be monitored for reactivation of infection. In HIV/HBV co-infected individuals, where OBI is frequent, higher viral loads have been detected in individuals infected with subgenotype A1 [30,88]. Although it was previously assumed that HBsAg-negative individuals were not infectious, reports show that blood and organ recipients were infected with HBV from donors who had no detectable HBsAg and low-level viremia [87,89]. A blood donation with as little as 32 subgenotype A1 HBV DNA copies per mL of plasma has been shown to be infectious [90].

Several studies have shown the high prevalence of OBI in HIV-infected SA adults when compared to uninfected individuals [26,91,92]. The prevalence of OBI in SA varies between study populations. One study among healthcare workers in SA reported 6.7% prevalence [25], while studies on HIV-positive patients reported 5.4% and 23% prevalence [26,27]. In a retrospective study conducted in SA, HIV-infected individuals on ART were found to be at high risk for reactivation of HBV because of OBI [93]. Several mechanisms may explain the OBI phenomenon, and these have been investigated using cell culture models and analysis of patients’ isolates [94,95,96,97]. Numerous southern African studies have focused on the contribution of HBV S gene mutations [93,98,99]. Several mutations, mainly within the immunogenic major hydrophilic region of the S gene, were identified. Some of these mutations are common across HBV genotypes A-H [97] and may enable HBsAg to escape detection.

## 5. Novel HBV Prevention and Treatment Strategies

The challenges to treatment and vaccination strategies highlighted above make it clear that novel vaccine and therapeutic approaches are required. Gene-based vaccines and therapies against viral infections have shown great promise in pre-clinical and clinical studies. Research in the Antiviral Gene Therapy Research Unit (AGTRU), South Africa, has put extensive efforts into designing nucleic acid-based therapies and vaccines to counter HBV infection (Table 2). From these studies several promising candidates have been identified (described below).

### 5.1. HBV Vaccines

Discovery of the Australia antigen, now known as the HBsAg, in 1965 [120,121] was particularly valuable for advancement of vaccines that comprise HBsAg or derivatives of this antigen. Abundance of the envelope protein in serum of HBV chronic carriers prompted initial investigation into use of extracts from individuals infected with the virus as vaccines [122,123]. Earliest vaccines, purified from plasma of HBV-infected people, were licensed in 1982 and consisted of 22 nm subviral particles made up of HBsAg [124]. These plasma-derived vaccines were successfully used to immunize hundreds of millions of people throughout the world, including SA. However, although found to be safe, the risk of transmitting pathogens and prions was a concern. This prompted production of recombinant HBsAg to use in vaccinations. To ensure appropriate eukaryotic post-translational modification with optimal epitope configuration of recombinant HBsAg, eukaryotic yeast cells (*Saccharomyces cerevisiae*) were used to produce the vaccine antigen [125,126]. These recombinant vaccines were first licensed in the late 1980s; in SA and other parts of the world, they rapidly replaced plasma-derived vaccines. Yeast-derived HBsAg remains the most widely used antigen in vaccines against HBV and safely induces very good protection. More than 95% of healthy infants, children, and young adults are protected from HBV infection following immunization [127,128]. Viral vaccine escape mutants are rare because of the compact HBV genome that has restricted sequence plasticity.

Vaccination programs were initially targeted to groups who were at high risk of HBV infection [124]. However, it became clear that targeting high risk individuals had little impact on the global burden of HBV-related disease. As a result, in 1991, the WHO recommended that HBV vaccination be globally included in the immunization programs of all countries by 1997 [129]. After adopting this recommendation, together with the significant support of the Global Alliance for Vaccination and Immunization (GAVI), significant inroads have been made into limiting the spread of HBV infection [124]. GAVI provided essential support to enable vaccination programs to be efficiently implemented in the poorest of countries. Impressively, by 2019, HBV vaccination was included in immunization programs of 97% of the world’s countries. The best documented example of the effectiveness of HBV vaccination was in Taiwan. In this country, the number of HBsAg-positive individuals younger than 20 years decreased from 9.8% in 1984 to 0.6% in 2004, and there has been an associated drop in HBV-related complications [130].

Typically, HBV vaccines are administered as a three-dose schedule [124,131]. Two priming doses, given one month apart, are followed by a booster six months after the initial dose. The WHO recommendation is that the first dose to be given within 12 hours of birth to diminish mother-to-child transmission. This mode of spread was initially thought to be unimportant in sub-Saharan Africa, but significant in east and southeast Asia and the western Pacific islands. Although horizontal transmission among toddlers and children is more significant, recent evidence indicates that perinatal spread during childbirth is important in Africa [132]. Although the WHO recommends the HBV birth dose vaccine, there has been slow implementation of HBV birth dose vaccination in SA and most of the other African countries. In addition to epidemiological considerations logistical difficulties with getting the vaccine to all newborns, especially when babies are not delivered at medical facilities [133], limit implementation of HBV vaccination at birth.

Although implementing recommendations of the WHO has resulted in impressive global HBV immunization [124], HBV infection continues to be a major global health problem [134]. Improving prophylaxis and addressing shortcomings of existing vaccines are a priority. Examples of limitations of current vaccines are diminished protection in individuals who have other diseases, such as chronic renal insufficiency and HIV-1 infection, and when vaccines are administered to individuals over 40 years of age [124,128]. The requirement for 3 doses of the vaccine is also a problem because it imposes a logistical burden, which may be particularly important in resource-constrained areas. Availability of vaccines that require only one dose would simplify compliance and make implementation of vaccination programs easier, especially in Africa. Administration of such vaccines at birth would also contribute significantly to better prevention of HBV infections.

Given that most HBV chronic carriers mount poor immune responses to the virus, and that currently licensed therapies rarely cure CHB, therapeutic vaccination is another interesting line of investigation. With the emergence of the COVID-19 pandemic, considerable effort has gone into developing new vaccine technologies. Important advances have been made, which may have relevance to vaccination against HBV. These include use of mRNA-containing vaccines and recombinant adenoviral vectors expressing the Spike protein of SARS-CoV-2 [135]. Efficient induction of humoral and T-cell human immune responses provide prophylaxis against COVID-19. Current research in the AGTRU is aimed at harnessing similar technology to prevent HBV infection. mRNAs encoding HBV antigens are being formulated in lipid nanoparticles for use as vaccines. Moreover, recombinant adenoviral vectors expressing proteins of HBV are being investigated as vaccines. Preclinical evaluation is currently in progress to assess usefulness of single vaccine doses and durability of immunity. Therapeutic enhancement of anti-HBV T-cell immune responses in chronic carriers, without toxic hepatocyte killing, is another potential benefit of these new approaches to vaccination.

### 5.2. Anti-HBV Gene Therapy

#### 5.2.1. Gene Silencing

Discovery of the RNA interference (RNAi) pathway heralded a new paradigm in programmable gene silencing [136]. The ease of repurposing the pathway, coupled with the impressive potency of gene knockdown, has led to silencing being lauded as a promising tool of gene therapy. The endogenous pathway is triggered by double-stranded RNA intermediates, viz. primary microRNAs (pri-miRNAs), precursor miRNAs (pre-miRNAs), and miRNA duplexes, to silence mRNA sharing complementary bases to these activators. Naturally, pri-miRNAs are processed by the RNAi machinery to pre-miRNAs, which in turn are processed to form the mature miRNA duplex. A guide strand is selected from the miRNA duplex after entering the RNA-induced silencing complex (RISC). This guide then directs the complex to partly complementary mRNA for silencing. By introducing mimics of these intermediates into the pathway, it is possible to reprogram RISC to silence any gene of interest. The potential for exploiting the pathway as a therapeutic modality was immediately obvious and a concerted effort was undertaken to evaluate RNAi for the inhibition of pathology-causing genes. HBV was no exception and researchers across the world [137,138,139,140,141,142,143] and from SA [114,144] explored silencing HBV gene expression using various RNAi activators (Figure 1). The AGTRU led the efforts on the continent to develop RNAi-based therapies.

Because CHB is a life-long affliction, it was logical to pursue a therapeutic strategy that allowed for long-term suppression of viral replication. Expression of RNAi activators from DNA cassettes allows sustainable expression of the therapeutic sequence and fulfils the requirement for long-term viral suppression. Initially developed gene silencing cassettes entailed expression of short hairpin RNA (shRNA) sequences [145], which comprise a double-stranded stem sequence joined with a single-stranded loop. As shRNAs are structurally similar to pre-miRNAs, they mimic these RNAi intermediates and are processed accordingly. Expressing anti-HBV shRNA sequences from DNA cassettes proved exceptionally efficacious as demonstrated by the AGTRU [114]. The authors targeted shRNAs to the *HBx* ORF of the HBV genome, reasoning that the presence of this sequence at the 3′ end of all viral transcripts would allow their simultaneous silencing with a single RNAi activator. This study was also one of the first to show expression of anti-HBV shRNA from a recombinant adenoviral (AdV) vector in a clinically relevant murine model of HBV replication. In vivo delivery of the shRNA-expressing DNA cassettes to the livers of transgenic mice effected 80–100% knockdown of viral gene expression, and inhibition was sustained for up to 28 days after administration of a single dose of the AdVs.

Advances in the field of RNAi led to development of improved gene silencers, which were based on structures of naturally occurring miRNA sequences. Artificial miRNAs, as these sequences came to be called, represented the next generation of expressed RNAi activators and exhibited improvements over expressed shRNAs. Artificial miRNAs are designed to resemble natural miRNAs, which comprise imperfectly matched hairpin structures, more closely. The AGTRU explored use of anti-HBV artificial pri-miRNAs (apri-miRNAs) based on the structures of naturally occurring pri-miR-31 and pri-miR-122 [113,146]. Anti-HBV shRNA sequences previously described by this group [114] were redesigned to mimic the secondary structure of pri-miR-31 and pri-miR-122. Furthermore, approximately 50 nucleotide sequences flanking natural pri-miR-31 and pri-miR-122 were included in the apri-miRNA. The result was highly efficient RNAi activators that achieved >80% knockdown of viral replication in cultured mammalian cells. As these activators mimicked pri-miRNA, which are typically transcribed by RNA polymerase (Pol) II, they could also be expressed from Pol II promoters. apri-miRNA cassettes are thus more versatile than their shRNA counterparts, which are limited to expression from Pol III promoters. The apri-miRNAs produced were processed efficiently and transcribed at much lower levels than shRNAs, which limits potentially toxic interference with the endogenous RNAi pathway. Saturation of the endogenous pathway from shRNA expression causes significant toxicity and death in mice [147]. To enable efficient delivery to hepatocytes in vivo, anti-HBV apri-miRNA-encoding cassettes were incorporated into AdVs and AAVs [36,37,119,148]. Delivery of the gene silencers with these vectors achieved potent (80–90%) and sustained (up to 8 weeks) gene silencing in mouse models of HBV replication.

In addition to employing DNA expression cassettes to produce RNAi activators, chemically synthesized short interfering RNAs (siRNAs) have also been explored by the AGTRU. Synthetic small interfering RNAs (siRNAs) typically contain ~21 perfectly matched base pair double strands with 2 nucleotide 3′ overhangs that mimic miRNA duplexes. After entering cells, siRNAs activate cytoplasmic RISC to reprogram the pathway. In contrast to expressed RNAi activators, siRNAs are functional over a short period. Although siRNAs have a short window period of silencing, this may still be useful for HBV therapy. The high viral antigenemia common in CHB is thought to suppress the immune response to the virus, allowing tolerance to the virus to build. By suppressing viral antigenemia, it is possible to restore the immune response and eliminate tolerance to the virus [149]. This concept has renewed interest in anti-HBV therapeutics that suppress viral antigenemia, even for a short period of time. To achieve such an effect with chemically synthesized siRNAs, it is imperative that the efficacy, safety, and stability of these molecules be optimized. In the case of synthetic siRNAs, this is invariably achieved through use of chemical modifications. The AGTRU in collaboration with research partners from the School of Chemistry at their host institute as well as the Rega Institute for Medical Research at the Katholieke Universiteit Leuven, Belgium described novel altritol-modified siRNAs targeted against HBV [117]. Altritol-modified nucleosides contain a 6-carbon sugar moiety as opposed to the naturally occurring 5-carbon ribose present in nucleosides. Altritol modification of the 3′ ends of the siRNAs was well tolerated and conferred favorable properties by reducing immunostimulation while maintaining silencing efficacy. In collaboration with German and French partners, the AGTRU also explored guanidinopropyl modification of chemically synthesized siRNA [115,116]. Addition of the guanidinopropyl group to the 2′-O of the ribose yielded modified siRNAs with increased stability, improved immune evasion, and reduced off-target effects. Importantly improvements in physicochemical properties of altritol- and guanidinopropyl-modified siRNA observed in cultured mammalian cells were also observed to observed in transgenic HBV mice.

RNAi remains an important tool for gene silencing and work in SA, using expressed and synthetic activators, demonstrates the potential of the technology for effectively treating chronic HBV infection.

#### 5.2.2. Gene Editing and Gene Modifiers

Drugs designed to disable or eliminate the episomal cccDNA could improve the likelihood of achieving a CHB cure. Current therapies focus on decreasing hepatitis B viremia by preventing new virion formation and secretion, or by blocking viral entry into hepatocytes. However, these approaches do not target the source of viral replication and persistence. Maintenance of the cccDNA as a minichromosome-like structure in long-lived hepatocytes can lead to HBV reactivation, stressing the importance of developing gene therapies that act directly on the cccDNA.

In SA, researchers have been exploring methods of disrupting, degrading, or silencing cccDNA using gene editing tools, such as designer nucleases and epigenetic modifiers. Transcription Activator-Like Effector Nucleases (TALENs) and Clustered Regularly Interspaced Short Palindromic Repeats (CRISPR) with CRISPR associated (Cas) RNA-guided nucleases have been used to cleave both integrated viral DNA and episomal cccDNA (Figure 1) [33,34,35,150]. TALENs are engineered nucleases created by fusing a TALE DNA binding domain to an endonuclease, typically derived from the catalytic region of the *Fok*I enzyme. To achieve targeted cleavage, TALENs are designed in pairs. Binding of a pair of TALENs to the pre-defined DNA sequence aligns the endonuclease domains to enable specific target cleavage and formation of a double-strand break (DSB). In the absence of a homologous donor sequence, these DSBs are repaired by error-prone non-homologous end joining (NHEJ) to result in insertions and deletions (indels). This approach can be used to disrupt the HBV genome permanently and reduce viral fitness [33]. CRISPR/Cas RNA-guided nucleases occur naturally in bacteria and archaea and serve as an adaptive immune system of these organisms. Type II CRISPR/Cas systems have been tailored to expedite gene editing strategies and are predominantly favored because of their simple design concept. A guide RNA with sequence complementarity to the target site binds and subsequently recruits the Cas endonuclease to this site. Action of the endonuclease leads to formation of a DSB. As with TALENs, errors introduced during NHEJ-mediated repair of the DSB disrupt the viral genome [34].

Heterodimeric TALENs targeting the HBV *surface*, *core*, and *polymerase* genes were first described by researchers at AGTRU in 2013 [33]. *Surface*- and *core*-targeting TALENs demonstrated the highest antiviral efficiency as both achieved targeted mutation of HBV DNA in vivo in a murine model of HBV replication. This effect was associated with significant reductions in viral replication markers, including >90% loss of circulating HBsAg and a three-fold decrease in intrahepatic HBcAg. Importantly, in vitro disruption of cccDNA was achieved with *surface*-TALENs at frequencies of up to 35%. Recently, second and third generation TALENs, which act as obligate heterodimers, have been described [35]. By incorporating pre-defined mutations in the *Fok*I endonuclease domains, heterodimeric but not homodimeric TALEN pairs can interact and cleave their cognates. This effectively reduces the likelihood of unintended off-target cleavage. Second-generation TALENs targeting HBV *surface* and *core* demonstrated similar in vitro and in vivo antiviral efficacies when compared to their first-generation heterodimeric counterparts. Importantly the second-generation gene editors demonstrated better specificity. Next-generation sequencing revealed a potential off target cleavage site in the intronic region of the *phenylalanine hydroxylase* gene when using the *core* TALENs. The percentage disruption at this unintended target was reduced when using the second-generation obligate heterodimers.

Anti-HBV CRISPR/Cas gene editing strategies using the *Staphylococcus aureus* Cas9 endonuclease (saCas9) have been shown to target, disrupt, and degrade integrated viral DNA and cccDNA. A single guide RNA targeting the *surface* ORF (sgRNA-8) inhibited viral replication across multiple HBV genomes (A1, A2 and D3). By using the smaller saCas9 endonuclease, Scott and colleagues were able to package the sgRNA-8 and saCas9 into a single stranded AAV (ssAAV) vector. Delivery of the ssAAV-SaCas9-sgRNA-8 to HBV-infected hNTCP-HepG2 cells caused a 55% reduction in secreted HBsAg levels and a 60% decrease in viral particle equivalents (VPEs), with accompanying decreases (up to 80%) of intracellular cccDNA and a four-fold loss of other HBV DNA forms. Interestingly episomal cccDNA was eradicated in this model of infection, suggesting complete degradation of the viral DNA. In HepG2.2.15 cells, which constitutively replicate HBV from integrated greater-than-genome length sequences, similar reductions in markers of viral replication were observed. Disabling indels were also observed in integrated HBV DNA [34]. Importantly, this gene therapy approach highlights usefulness of the strategy to inactivate viral replication at the source.

Targeted epigenetic engineering approaches have also demonstrated anti-HBV efficacy and could be used to silence the cccDNA permanently [151]. Because viral and endogenous gene transcription is similarly regulated by host–cell pathways, the minichromosome-like structure of the cccDNA is amenable to epigenetic regulation. To test whether targeted epigenetic modifications could impede viral replication, repressor TALEs (rTALEs) comprising a TALE DNA binding domain and Krüppel-associated box (KRAB) repression domain were designed to bind to the *surface* open reading frame of HBV [152]. Reductions in HBsAg (80–95%), viral mRNA (50–90%), and circulating VPEs (50–75%) were observed in a murine model of HBV replication. These reductions correlated with a 50% increase in methylation at the naturally occurring CpG island II of the HBV genome, specifically at CpG position 38. In a clinical setting, methylation of HBV cccDNA at CpG island II was associated with HBeAg-negative patients, suggesting increased methylation may inhibit viral replication [153,154].

#### 5.2.3. Gene Delivery

Research aimed at designing nucleic acid-based anti-HBV therapies has progressed and promising candidates have been identified. However, challenges of delivering these nucleic acids efficiently to target cells in vivo hampers progression to clinical evaluation. Liver-targeted delivery systems in the form of non-viral vectors (NVVs) and viral vectors (VVs) have been developed and serve as valuable tools for gene therapy against HBV (Figure 1). Cationic lipids and polymers are the most widely used NVVs. However, lack of tissue specificity and the ability to form aggregates with serum proteins reduce transfection efficiencies. Ease of functionalization overcomes some of these challenges. Studies showing that NVV lipids or polymers can be targeted to the liver when linked to asialoglycoprotein receptor (ASG-R, abundant on hepatocyte surface) ligands has given momentum to the development of liver specific NVVs. Research in the Department of Biochemistry, Non-Viral Gene Delivery Laboratory at the University of KwaZulu-Natal, has designed and tested several ASG-R-targeted, highly stable, and safe NVVs. These include functionalized liposomes and gold/selenium nanoparticles [155,156,157,158]. Recently designed lactobionic acid (ASG-R ligand) gold/selenium nanoparticles resulted in improved transgene expression, high DNA stability, and lower toxicity in liver-derived cell lines [158,159,160]. The demonstration that synthetic nucleic acids, such as siRNAs, can be delivered with these ASG-R-targeting liposomes is of significance to anti-HBV siRNA-based gene therapy [161]. Most important is the demonstration by collaborating SA teams that cationic lipids conjugated to galactose and stabilizing polyethylene glycol (PEG) and 1,2-dioleoyl-sn-glycero-3-phosphoethanolamine (DOPE) as a helper lipid efficiently deliver HBV targeting siRNAs to liver-derived cell lines. The formulations also effectively inhibited HBV gene expression in vivo without obvious adverse effects [162,163,164].

Unlike siRNAs, cassettes encoding nucleic acid therapeutics, such as shRNA, apri-miRs, TALENs, and CRISPR/Cas sequences, should provide a sustained therapeutic effect and are compatible with delivery using VVs. The high efficiency of gene transfer and persistence for prolonged periods confer advantages on these genetically modified viruses for nucleic acid-based therapy of CHB. HBV research has benefitted extensively from the development of AdVs, lentiviral vectors (LVs), and AAVs. The application of AdVs, LVs, and AAVs to deliver anti-HBV gene therapeutics has been extensively studied in the AGTRU. The initial study engineered first generation AdVs, which still bear most of the viral genes, to express HBV-targeting shRNAs or apri-miRs. These studies showed profound liver-specific gene expression and reduction of HBV replication markers, albeit with a strong AdV-induced immunity with short-term therapeutic effects [114]. Modification of these AdVs with PEG reduced the inflammatory response, adaptive immune response, toxicity, and prolonged therapeutic effects after a second dose [118]. To further reduce the immune response activated by viral gene expression, later studies used gutless or helper-dependent adenoviral vectors (HdAdVs) to express shRNAs or apri-miRs. These studies demonstrated diminished immune stimulation by the vectors and prolonged therapeutic effects [37,119]. Feasibility of using LVs for HBV-targeted gene therapy has also been demonstrated [165]. Significant knockdown of HBV gene expression without obvious adverse effects was observed after administering antiviral LVs to neonatal HBV transgenic mice. The diminished HBV replication markers were demonstrated for the study period of one year, which is approximately the normal lifespan of a mouse.

As a result of their favorable biosafety profile, AAVs are popular VVs and are well-suited for HBV gene therapy [166]. The first studies using AAVs in the AGTRU delivered apri-miR or CRISPR/Cas sequences to cultured cells and/or mice. An AAV8 vector expressing apri-miR suppressed HBV gene expression in vitro and in vivo. In vivo efficacy lasted for about ten months without obvious toxicity [36]. AAV2 vectors expressing Cas9 and an HBV-specific guide RNA resulted in reduction of HBV gene expression by up to 95%. Importantly, 61% of on-target indels and only 0.05% of off-target indels were detected [34]. Current studies using a synthetic ancestral AAV (Anc80) to deliver anti-HBV apri-miR-encoding cassettes show similar efficacy to that observed with AAV8 [167].

## 6. Discussion

Subgenotype A1, the predominant strain circulating in SA, has been extensively characterized and shown to have unique molecular characteristics with a high hepatocarcinogenic potential. As we progress towards meeting the targets of HBV elimination, it is important that there is an awareness of the uniqueness of subgenotype A1. This is essential to diagnose infection and design appropriate treatment strategies. Mutations identified in HBV from HCC patients may provide cost-effective biomarkers for prioritizing HBV-infected patients for treatment. This is important in sub-Saharan Africa, where there are both resource and capacity limitations. Moreover, it is imperative that individuals infected with subgenotype A1 are well represented in clinical trials aimed at evaluating new antiviral modalities.

Increased HBV replication in HBV/HIV co-infected mothers is likely to increase perinatal HBV transmission. Studies have shown that HIV co-infection increases carriage of HBeAg and increases HBV replication in pregnant women [105,106]. This poses an increased risk of perinatal transmission of HBV from infected mothers, which is especially problematic when diagnosis of HIV is not timeous, and ART not provided [168]. This risk can be mitigated by antenatal testing for both HIV and HBV infection. Antenatal screening of HIV is already performed in the public sector and analysis has been performed showing that the same can be done cost-effectively for HBV [169]. Furthermore, a study from the Western Cape province of SA showed that pregnant women are amenable to HBV screening [170]. Testing could be enhanced by using combination rapid tests for simultaneous diagnosis of both HBV and HIV infections. Such combination rapid tests are already available, although not yet pre-approved by the WHO [171]. Through screening and identifying HBV-infected pregnant women, TDF-based therapy can be used to decrease the likelihood of perinatal transmission.

SA has not yet implemented a birth dose vaccine against HBV, despite recommendations of the National Guidelines for the Management of Viral Hepatitis published in 2019 [78]. The rationale behind the old but currently used schedule is that the first dose of vaccine administered at six weeks protects the neonate by preventing subsequent horizontal HBV transmission in infancy which historically has been the usual route of infant transmission in SA. Risk of perinatal mother to child transmission of HBV during the first six weeks of life is low if HBV replication is low in the infected mother. It is also thought that passively transferred maternal antibodies protect infants before making their own antibodies in response to HBV immunization [172]. Strategies using viral vectors and mRNA technology to develop anti-viral vaccines have shown promise [173,174,175]. Developments in these novel technologies may well influence vaccination strategies in SA and other parts of the world.

Failure of the current therapies to eliminate cccDNA calls for novel strategies to clear HBV infection. Newly designed gene editors, such as TALENs and CRISPR/Cas, may eradicate cccDNA and offer a potential for sterilizing cure. However, progress of these therapies to clinical testing has been delayed. Finding safe and efficient delivery methods are essential for clinical application of the technology. Considerable research efforts are going into optimizing liver-targeted NVVs and VVs, which augurs well for achieving the goal of eradicating HBV infection. Such efforts should be supported, initiated, and implemented in sub-Saharan Africa, where the need is greatest, to ensure human and infrastructural capacity development, independence, and sustainability.

Research in HBV and HBV/HIV infection epidemiology has made significant progress and is well poised to contribute to SA’s guidelines and improve hepatitis B management. Although results from using gene therapy against HBV are very encouraging, a few hurdles need to be overcome before clinical testing (Table 2). Access to physiologically relevant HBV infection models such as the chimpanzee model is made difficult by ethical concerns and/or high cost. Transgenic mice and murine hydrodynamic injection models may be used to simulate HBV replication in vivo. However, all stages of the infection cycle, notably entry of the virus into hepatocytes and formation of cccDNA, are not reproduced. Viral vector-mediated prolonged expression of gene editors in the host may also be associated with off-target effects and toxicities in a clinical setting. Hence, current efforts are focusing on developing HBV models that will resemble the entire HBV replication cycle and NVVs to deliver gene editor-encoding mRNA.

## Figures and Tables

**Figure 1 viruses-14-01939-f001:**
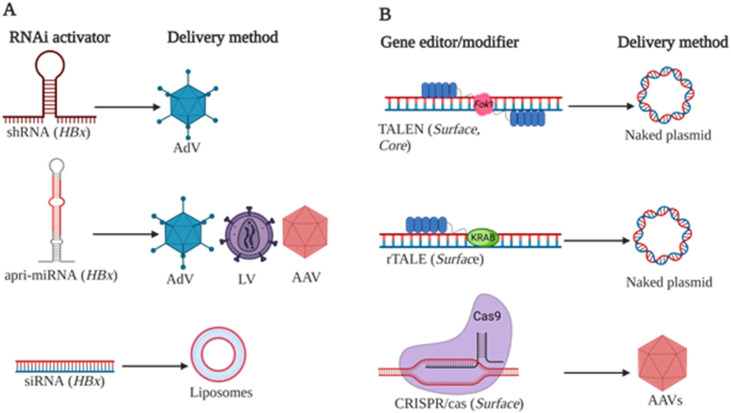
Gene therapy tools used for targeted disruption of HBV replication in South Africa. (**A**) RNAi activators such as short hairpin RNAs (shRNAs), artificial primary microRNAs (apri-miRNAs), and short interfering RNAs (siRNAs) targeting hepatitis B *x* (*HBx)* sequence were designed and delivered using adenoviral vectors (AdVs), AdVs/lentiviral vectors (LVs)/adeno-associated viral vectors (AAVs) and liposomes, respectively. (**B**) Transcription activator-like effector endonucleases (TALENS) targeting *Core* or *Surface* and Transcription activator-like repressors (rTALEs) targeting *Surface* were generated and delivered as naked plasmid DNA. Clustered Regularly Interspaced Short Palindromic Repeats (CRISPR) with CRISPR-associated (Cas) RNA-guided nucleases (CRISPR/c as) targeting *Surface* were designed and delivered using AAVs (Created with biorender.com).

**Table 1 viruses-14-01939-t001:** Molecular and functional characterization of HBV subgenotype A1 variants.

Target	Mutation	Major Findings	References
BCP/Pre-Core	G1888A	Interfere with initiation at the core AUG and decrease core protein translation.	[46]
	A1762T/G1764A	Results in decreased HBeAg transcription. Risk factor for hepatocellular carcinoma (HCC) development. Occurs more frequently in HBV/HIV co-infected patients.	[47,48,49,50,51,52,54]
	GCAC to TCAT at 1809-1812	Affects translation of HBeAg by a leaky ribosomal scanning mechanism.	[48,53,55]
	G1862T	Affects HBeAg expression at the post-translational level. Frequent in HBeAg-negative South African carriers and in HCC tumorous liver tissue. Interferes with the maturation of the precursor to HBeAg. Reduces HBeAg secretion. Leads to the accumulation of the HBeAg precursor protein in the endoplasmic reticulum (ER) and endoplasmic reticulum Golgi intermediate compartment (ERGIC), leading to increased ER stress.	[45,47,48,56,57,58,59,60]
Pre-S	Pre-S2 deletion	Occurmore frequently in patients with HCC. Mediate immune escape. Occur more frequently in HBV/HIV co-infected patients.	[50,54,61,62]
	Pre-S2 initiation codon mutation		
	ps2F22L		

**Table 2 viruses-14-01939-t002:** Key areas of HBV research in South Africa.

Research Group/s	Research Area	Research Field	Key References
Hepatitis Diversity Research Unit, Department of Internal Medicine, School of Clinical Medicine, Faculty of Health Sciences, University of the Witwatersrand, Johannesburg	- Epidemiology of HBV	Clinical	[71,100]
	- Molecular and functional characterization of HBV	Clinical	[46,50,54,60,101,102]
	- HBV/HIV co-infection	Clinical	[31,103]
	- Occult HBV infection	Clinical	[104]
Division of Medical Virology, Faculty of Medicine and Health Sciences, Stellenbosch University, Cape Town	- HBV and HBV/HIV infection epidemiology	Clinical	[105,106,107,108,109]
	- HBV and HBV/HIV infection management	Clinical	[72,80,83,110]
HIV and Hepatitis Research Unit, Department of Virology, Sefako Makgatho Health Sciences University, Pretoria	- HBV and HBV/HIV infection epidemiology	Clinical	[26]
	- HBV and HBV/HIV infection management	Clinical	[29,111,112]
	- Occult HBV infection	Clinical	[91]
Wits/SAMRC Antiviral Gene Therapy Research Unit, IDORI, Faculty of Health Sciences, University of the Witwatersrand, Johannesburg	- Anti-HBV vaccine development	Pre-clinical	Not published
	- Anti-HBV gene therapy development	Pre-clinical	[33,34,35,113,114,115]
	- Non-viral vector anti-HBV gene delivery	Pre-clinical	[116,117]
	- Viral vector anti-HBV gene delivery	Pre-clinical	[36,37,118,119]

## Data Availability

Not applicable.

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
