# Peer review of "Hepatitis B Virus Research in South Africa"

_viruses, 2022, doi:10.3390/v14091939_

Round 1

Reviewer 1 Report

The authors wrote a comprehensive review with the title ‘Hepatitis B virus research in South Africa’ about HBV infection, with focus on the subgenotype A1, which is prevailing in South Africa (SA) and described in this manuscript the situation about HIV/HBV coinfected and occult infected individuals in SA. In addition, novel HBV prevention and treatment strategies, including anti-HBV gene therapy were summarized.

The paper is clearly written. However, there is a break in the reading flow in chapter 5.2, which almost gives the impression that the section before and after were written by different people. From 5.2 on, the level of detail is much greater than in the previous sections.

Major points:

11)   The authors mentioned in the last sentence of their abstract: ‘The review summarizes research carried out in SA….’. It is not always clear which part is a general overview of HBV and which is specific to SA. The authors should revise their manuscript and make this clearer in some places.

22)      There are many references missing, for example line 40-63, 107, 328-336, 380.

33)      The authors described that almost only HBV/HIV co-infected patients receive indirect treatment for HBV with TDF, which is also be used to treat HIV infection. It would be beneficial if the authors would include a section on whether patients infected with HBV alone receive treatment and, if so, when and for how long.

44)      There is no real connection between the part from 5.2 and the parts before. It is difficult for the reader to follow the text. Especially this part starts directly with many technical details of gene silencing without reference to the topic HBV.

55)      The authors describe starting 5.2 the anti-HBV gene therapy approaches performed mainly in SA.

a.       Most of the papers summarized in this review are be published by the group of Patrick Arbuthnot. In mine opinion, this is ok. However, this should be state clear in this manuscript not washed out by statements like ’by a group from SA’ (line 373) or Ely and colleagues (line 386).

b.       Could the authors indicate for me, which other research groups in SA contributed to topic described in starting in section 5.2?

c.       The authors need to make clearer if the experiments starting 5.2 are done in mouse or in vitro.

66)      The sentence in line 90/91 (‘Novel HBV prevention…’) is not connected to the section before.

Reviewer 2 Report

The authors provide a comprehensive overview of HBV research in South Africa, focusing on the A1 subgenotype and efforts to understand its pathogenesis and potential therapeutic avenues. In addition, the authors discuss co-infections such as HIV, and the impact upon HBV infection in South African individuals. Overall, the review is comprehensive, clearly written and informative.

Reviewer 3 Report

It is a pleasure to review the article entitled “Hepatitis B virus research in South Africa” authored by Maepa et al..  This manuscript was well-written and provided a comprehensive overview of the epidemiology and molecular biologies of HBV research.  The background information is sufficient enough for general readers and the references are precise.  I am providing a few discretionary suggestions/comments for your consideration:

 1.       I suggest including a small paragraph or table to showcase the research network in SA and key areas of HBV research in the institutes to increase visibility and potential collaborations. The current article mainly focused on nucleic acid therapy, is there any small molecule research in SA?  Has any of the research strategies progressed to clinical development

2.       Regarding HBV management, what are the criteria for stopping therapy, and how long is the treatment?  On line 113, you mentioned that 37% of the lamivudine treatment-naïve patients have drug resistance mutations.  Is this due to the transmitting resistant HBV or natural polymorphisms of the WT HBV?  Are patients with occult HBV infection being treated under the SA treatment guidelines?

3.       Figure 1.  I suggest adding “in South Africa” in the title.

4.       For both the RNAi and gene editing technologies, I suggest describing the results more specifically rather than using terms like  “significant reductions”,  “significant knockdown”, “highest efficiency”, “the percentage..was reduced” etc.

5.       As research is a work in progress, it might be a good idea to outline the challenges and future research directions to provide a balanced view.  This can be achieved in the Discussion Section.

6.       Zine-finger nucleases (ZFNs) (line 525); "otent" (line 401)???
